# Dysregulation of the PRUNE2/PCA3 genetic axis in human prostate cancer: from experimental discovery to validation in two independent patient cohorts

Richard C Lauer[1,2†], Marc Barry[3†], Tracey L Smith[4,5], Andrew Maltez Thomas[6], Jin Wu[1,7], Ruofei Du[8], Ji-Hyun Lee[9,10], Arpit Rao[11], Andrey S Dobroff[1,12], Marco A Arap[13,14], Diana N Nunes[15], Israel T Silva[16], Emmanuel Dias-Neto[15], Isan Chen[17], Dennis J McCance[1,7], Webster K Cavenee[18], Renata Pasqualini[4,5]*, Wadih Arap[4,19]*

[1]University of New Mexico Comprehensive Cancer Center, Albuquerque, New Mexico, United States; [2]Division of Hematology/Oncology, Department of Internal Medicine, University of New Mexico School of Medicine, Albuquerque, New Mexico, United States; [3]Department of Pathology, University of Utah, Salt Lake City, Utah, United States; [4]Rutgers Cancer Institute of New Jersey, Newark, New Jersey, United States; [5]Division of Cancer Biology, Department of Radiation Oncology, Rutgers New Jersey Medical School, Newark, New Jersey, United States; [6]Department of Biochemistry, Institute of Chemistry, University of São Paulo, São Paulo, Brazil; [7]Department of Pathology, University of New Mexico, Albuquerque, New Mexico, United States; [8]Department of Biostatistics, University of Arkansas for Medical Sciences, Little Rock, Arkansas, United States; [9]Department of Biostatistics, University of Florida, Gainesville, Florida, United States; [10]Division of Quantitative Sciences, University of Florida Health Cancer Center, Gainesville, Florida, United States; [11]Section of Hematology and Oncology, Department of Medicine, Baylor College of Medicine, Houston, Texas, United States; [12]Division of Molecular Medicine, Department of Medicine, Albuquerque, United States; [13]Division of Urology, University of São Paulo Medical School, São Paulo, Brazil; [14]Syrian-Lebanese Hospital, São Paulo, Brazil; [15]Laboratory of Medical Genomics, A.C. Camargo Cancer Center, São Paulo, Brazil; [16]Laboratory of Bioinformatics and Computational Biology, A.C. Camargo Cancer Center, São Paulo, Brazil; [17]MBrace Therapeutics, San Diego, California, United States; [18]Ludwig Institute for Cancer Research, University of California, San Diego, La Jolla, California, United States; [19]Division of Hematology/Oncology, Department of Medicine, Rutgers New Jersey Medical School, Newark, New Jersey, United States

*For correspondence:
renata.pasqualini@rutgers.edu (RP);
wadih.arap@rutgers.edu (WA)

†These authors contributed equally to this work

## Abstract

**Background:** We have previously shown that the long non-coding (lnc)RNA *prostate cancer associated 3* (*PCA3*; formerly *prostate cancer antigen 3*) functions as a trans-dominant negative oncogene by targeting the previously unrecognized prostate cancer suppressor gene *PRUNE2* (a homolog of the *Drosophila prune* gene), thereby forming a functional unit within a unique allelic locus in human cells. Here, we investigated the *PCA3/PRUNE2* regulatory axis from early (tumorigenic) to late (biochemical recurrence) genetic events during human prostate cancer progression.

**Methods:** The reciprocal *PCA3* and *PRUNE2* gene expression relationship in paired prostate cancer and adjacent normal prostate was analyzed in two independent retrospective cohorts of clinically annotated cases post-radical prostatectomy: a single-institutional discovery cohort (n=107) and a multi-institutional validation cohort (n=497). We compared the tumor gene expression of *PCA3* and *PRUNE2* to their corresponding expression in the normal prostate. We also serially examined clinical/pathological variables including time to disease recurrence.

**Results:** We consistently observed increased expression of *PCA3* and decreased expression of *PRUNE2* in prostate cancer compared with the adjacent normal prostate across all tumor grades and stages. However, there was no association between the relative gene expression levels of *PCA3* or *PRUNE2* and time to disease recurrence, independent of tumor grades and stages.

**Conclusions:** We concluded that upregulation of the lncRNA *PCA3* and targeted downregulation of the protein-coding *PRUNE2* gene in prostate cancer could be early (rather than late) molecular events in the progression of human prostate tumorigenesis but are not associated with biochemical recurrence. Further studies of PCA3/PRUNE2 dysregulation are warranted.

**Funding:** We received support from the Human Tissue Repository and Tissue Analysis Shared Resource from the Department of Pathology of the University of New Mexico School of Medicine and a pilot award from the University of New Mexico Comprehensive Cancer Center. RP and WA were supported by awards from the Levy-Longenbaugh Donor-Advised Fund and the Prostate Cancer Foundation. EDN reports research fellowship support from the Brazilian National Council for Scientific and Technological Development (CNPq), Brazil, and the Associação Beneficente Alzira Denise Hertzog Silva (ABADHS), Brazil. This work has been funded in part by the NCI Cancer Center Support Grants (CCSG; P30) to the University of New Mexico Comprehensive Cancer Center (CA118100) and the Rutgers Cancer Institute of New Jersey (CA072720).

## Editor's evaluation

In this manuscript, the upregulation of PCA3 and downregulation of PRUNE2 in prostate cancer as compared with normal prostate were first discovered. The conclusions are reached through two large-scale clinical-derived cohorts, supporting that PCA3 and PRUNE2 function as an oncogene and a tumor suppressor gene, respectively. The major findings opened up a novel avenue for prostate cancer treatment.

## Introduction

Prostate cancer is the most common cancer and the second most common cause of cancer death in men (*Siegel et al., 2021*), and there continues to be a pressing need for new diagnostic and therapeutic approaches for this disease, as well as better prognostic biomarkers to guide treatment. Long non-coding RNA (lncRNA) species are increasingly recognized as having regulatory functions in tumorigenesis, and nucleic acid-based therapeutics are being developed as a promising means of targeting pathogenic lncRNAs (*Arun et al., 2018*). Several lncRNAs have recently been found to associate with prostate cancer, and the best known of these, *prostate cancer associated 3* (*PCA3*; formerly *prostate cancer antigen 3*) has been used clinically for many years as the most specific diagnostic biomarker for prostate cancer (*Bussemakers et al., 1999*; *de Kok et al., 2002*); however, its prognostic significance remains uncertain. Strikingly, *PCA3* emerged first only in mammals, with further evolution in primates (*Clarke et al., 2009*), and, given aspects of the sequence and genomic organization, we have hypothesized that it might have been introduced into the genome by an ancient oncogenic virus (*Teixeira et al., 2017*). In humans, *PCA3* has an unusual genomic organization, being present in an antisense direction within an intron of the protein-coding gene *PRUNE2*. Somewhat surprisingly for a molecule that is well established as a Food and Drug Administration (FDA)- and European Medical Agency (EMA)-approved biomarker, relatively little was known about the biological function of *PCA3* until recently. *Ferreira et al., 2012*, showed that *PCA3* is androgen-regulated and that it promotes prostate cancer cell survival. Subsequently, we have established that *PCA3* downregulates the expression of *PRUNE2* in a rather unusual way: at the RNA level by RNA editing mediated via adenosine deaminase RNA-specific family members (*Salameh et al., 2015*). We have shown that expressing ectopic *PCA3* or, alternatively, silencing *PRUNE2* induced cell transformation and cell proliferation in vitro,

increased adhesion and migration of prostate cancer cells, and yielded larger tumors in xenograft tumor models. The opposite biological effects were seen with *PCA3* silencing or ectopic *PRUNE2* expression (*Salameh et al., 2015*). Preliminary studies of human prostate cancer samples compared to normal prostate showed increased *PCA3* expression, decreased *PRUNE2* expression, and evidence for RNA editing of these genes. Based on these experimental findings, we proposed that there is a functional molecular axis in human prostate cancer in which *PCA3* acts as a transdominant-negative oncogene to downregulate a previously unrecognized tumor suppressor gene, *PRUNE2* (*Salameh et al., 2015*).

Here, we propose that this molecular interplay may serve as a translational target for diagnostic and/or therapeutic intervention in human prostate cancer. First, we present additional correlative evidence from two retrospective post-surgical primary prostate cancer cohorts in support of our experimental model of *PCA3* as a dominant-negative oncogene and *PRUNE2* as a tumor suppressor gene and for their co-regulation in human prostate cancer. Moreover, we examine the dysregulation of the *PCA3*/*PRUNE2* regulatory axis across tumors of different grades (patterns), stages, and groups (*Gordetsky and Epstein, 2016*; *van Leenders et al., 2020*). Finally, we assess whether tumor expression levels of *PCA3* and/or *PRUNE2* are prognostic of biochemical disease recurrence after surgery.

## Methods

### Discovery patient cohort

Based on a power analysis using gene expression data from our prior work (*Salameh et al., 2015*), for the UNMCCC single-institutional discovery cohort, we searched the archives of the Department of Pathology at the UNM School of Medicine for at least 100 consecutive patients (final cohort size: n=107) who had a radical prostatectomy as the primary treatment for organ-confined prostate cancer between the years 2001 and 2013 and who had the following clinical and pathological attributes: final post-prostatectomy Gleason Score 7 (either Gleason Grade Group 2 (3+4) or Gleason Grade Group 3 (4+3)), pathological stage pT2 or pT3a, negative surgical margins, negative for seminal vesicle invasion, no evidence of local or distant metastasis, and no prior treatment for prostate cancer. The following additional data were retrospectively abstracted from the individual medical records: age at surgery, race, presence of recurrence, type of recurrence (i.e., biochemical, local, metastatic), and disease-free survival time. Biochemical disease recurrence was defined as a detectable serum prostate-specific antigen concentration of at least 0.2 ng/ml post-operatively. Lost to follow up was defined as not having been followed up at the UNMCCC after their urological surgery. All included cases had an independent pathological re-review by a Board-certified pathologist with expertise in urological pathology (MB), with confirmation of diagnosis, Gleason-based analysis (grading, scoring, and grouping), standard TNM staging, and margin status post-resection. A small number of identified cases (<5%) had to be excluded due to the very limited amount of tumor present.

### Microdissection of tumor and normal prostate (nonneoplastic prostatic glandular tissue) for the discovery cohort

To obtain tumor for RNA analysis, a representative carcinoma-containing formalin-fixed paraffin embedded (FFPE) block was chosen from each case. Contiguous foci of tumor were marked on the glass slide such that the density of tumor cells was at least 75%. The boundary of the corresponding areas on the tumor block was scored with a blade tip, effectively allowing microdissection of tumor in the process of microtome sectioning. Multiple 10 μm sections were cut, depending on the area of the tumor focus/foci. In 24 (22.4%) of the cases, we also microdissected areas of nonneoplastic prostatic glandular tissue away from tumor in a similar manner, again also aiming for at least 75% epithelial density.

### Measurement of PRUNE2 and PCA3 gene expression in the discovery cohort by quantitative RT-PCR

Briefly, gene expression for *PCA3* and *PRUNE2* were determined by quantitative reverse transcription polymerase chain reaction (qRT-PCR) by using TaqMan gene expression assays (Thermo Fisher Scientific) with amplicon detection via a LightCycler 96 (Roche Diagnostics). Gene expression was quantified by the relative logarithmic RT-PCR threshold cycles (ΔCt) between the target genes and

housekeeping control genes (*Livak and Schmittgen, 2001*). Specifically, total RNA was extracted from the microdissected FFPE sections using the PureLink FFPE Total RNA Isolation Kit (Thermo Fisher Scientific, Cat. No. K1560-02). RNA was quantified on a NanoDrop ND-1000 Spectrophotometer (Thermo Fisher Scientific), and the average A260/A280 ratio was 1.94 (range 1.88–2.07), indicating optimal quality of the RNA extracted for gene expression assays. RNA was then further quantified with the Qubit RNA HS Assay Kit (Thermo Fisher Scientific, Cat. No. Q32852) on a Qubit 2.0 (Thermo Fisher Scientific) for accurate RNA concentration. RNA integrity was evaluated with the Agilent RNA 6000 Nano kit (Agilent Technologies, Cat. No. 5067-1511) on an Agilent 2100 Bioanalyzer (Agilent Technologies). To remove genomic DNA contamination, RNA samples were treated with 2 U of DNase I (Thermo Fisher Scientific, Cat. No. 18068-015) per 2 μg of total RNA. All procedures were performed according to the manufacturer's standard protocols.

Reverse transcription was performed in triplicate in order to create enough cDNA for the entire project. Five-hundred ng RNA in each of three tubes was reverse transcribed with the High-Capacity RNA-to-cDNA Kit (Thermo Fisher Scientific, Cat. No. 4387406) in a final volume of 20 μl, according to the manufacturer's instructions. Reverse transcription was carried out in a Gene Amp PCR System 9700 (Applied Biosystems) at 37°C for 60 min and terminated by 95°C for 5 min. Then, three aliquots were combined for the following experiments.

For the Thermo Fisher Scientific TaqMan gene expression assay experiments, three (Hs00322421_m1, Hs00999960_m1, and Hs01060890_m1) and two (Hs01371939_g1 and Hs03462121_m1) assays were chosen for target genes *PRUNE2* and *PCA3,* respectively (designated PR1, PR2, and PR3, and PC1 and PC2). Three endogenous controls GAPDH (Hs02758991_g1), HPRT1 (Hs02800695_m1), and UBC (Hs01871556_s1) were selected (designated C1, C2, and C3) (*Vandesompele et al., 2002*). Each *PRUNE2* assay and *PCA3* assay was labeled with FAM and paired with a VIC-labeled endogenous control in a duplex reaction, with separate reactions to include all of the three endogenous controls. Therefore, a total of fifteen duplex gene expression mixes, nine for *PRUNE2* and six for *PCA3*, was required for all specimens (*Figure 1—source data 1* [tumor] and *Figure 1—source data 1* [normal]).

Each duplex gene expression assay was then performed in triplicate for all specimens following the manufacturer's standard protocols, for a total of 45 expression measures for each case. qRT-PCR was performed with the TaqMan Gene Expression Master Mix (Thermo Fisher Scientific, Cat. No. 4369514) using 1 μl of each TaqMan target gene assay (20× FAM) and endogenous controls assay (20× VIC), 1 μl of cDNA template (equivalent to 25 ng RNA input), and 7 μl of RNase-free water for a 20 μl final reaction mixture. A non-template control was included in every master mix in every 96-format tray. In addition, in order to evaluate inter-plate variation, we also included one RNA sample, in triplicate, in all the 96-format trays. Analysis of these controls indicated that there were no significant batch effects (data not shown). The qRT-PCR product detection was achieved on a LightCycler 96 (Roche Diagnostics). The cycle program was: at 95°C for 10 min, followed by 40 cycles at 95°C for 15 s and at 60°C for 1 min. Quantification of target and control genes (Cq) in each sample was performed by LightCycler 96 SW 1.1 (Roche Diagnostics).

## Validation patient cohort

For The Cancer Genome Atlas (TCGA) patient validation cohort (n=497 patients), we first downloaded clinical data along with the expression of the lncRNA *PCA3* and the *PRUNE2* gene (http://cancergenome.nih.gov) with the UCSC Xena browser (*Cancer Genome Atlas Research Network, 2015*; *University of North Carolina TCGA Genome Characterization Center, 2017*), together with paired nonneoplastic samples in 52 of the cases (10.5%). The following clinical and pathological characteristics were included in the study: age at diagnosis, vital status, tumor Gleason-based analysis (grading, scoring, grouping), pathological stage, status of biochemical recurrence, and time to recurrence. Gene expression was calculated with $\log_2$RNA-Seq by Expectation-Maximization (RSEM) (*Li and Dewey, 2011*; *Goldman et al., 2020*). By using the available dataset, we evaluated *PCA3* and *PRUNE2* gene expression values in terms of tumor versus nonneoplastic prostate, biochemical recurrence, pathological T stage, Gleason analysis (grade, score, and group), and age at pathology-proven diagnosis. Because the regulation of *PRUNE2* by *PCA3* occurs at the RNA level by the formation of an RNA hetero-duplex, we also evaluated the ratio of the expression of the two genes in terms of the clinical and pathological variables for each patient of the cohort.

## Statistics

Demographic and clinical variables were summarized with descriptive statistics. For the discovery cohort, the mean and median of gene expressions across multiple control genes and assays were summarized, and these were used as measures for gene expression of *PRUNE2* and *PCA3* relative to endogenous housekeeping controls for each case. More detailed methods are described in Appendix 1.

Testing for differences of *PCA3* and *PRUNE2* expression between paired tumor and nonneoplastic prostate expression was by the Wilcoxon signed rank test. The Kruskal-Wallis test was used when comparing three or more groups. Assessment for significant differences of gene expression by recurrence status was by Wilcoxon rank sum test. The Kaplan-Meier product limit method with log-rank test was used to explore the relationship between gene expression levels or the ratio and the time to recurrence. Multivariable Cox proportional hazard modeling was used to fit for the association between time to recurrence and expression levels of *PRUNE2* or *PCA3* or their ratio, while controlling for multiple clinical covariates. All statistical analyses were carried out by using the SAS (9.4) or R software package (R 3.4.5), unless otherwise indicated (R and SAS codes are available in the *Source code 1*).

**Table 1.** Discovery cohort: clinicopathological features of the 107patients in this study who had radical prostatectomy at UNM for localized prostatic adenocarcinoma (further details in *Table 1—source data 1*).

| Variable | Number | % or Mean (STD; range) |
|---|---|---|
| Age (years) | 107 | 62.8 (8.4; 45–84) |
| | | |
| Race | | |
| Non-Hispanic White | 91 | 85 |
| Hispanic | 8 | 7.5 |
| American Indian | 3 | 2.8 |
| African American | 3 | 2.8 |
| Other | 2 | 1.9 |
| | | |
| Post-prostatectomy Gleason Grade (Grade Group) | | |
| 3+4 = 7 (Grade Group 2) | 93 | 86.9 |
| 4+3 = 7 (Grade Group 3) | 14 | 13.1 |
| | | |
| Pathological stage | | |
| pT2 | 80 | 74.8 |
| pT3a | 27 | 25.2 |
| | | |
| Biochemical recurrence status | | |
| No | 83 | 77.6 |
| Yes | 19 | 17.8 |
| LTF | 5 | 4.7 |

STD = standard deviation. LTF = lost to follow up.

The online version of this article includes the following source data for table 1:

**Source data 1.** Discovery cohort.

### Study approval

For the discovery cohort, there was University of New Mexico Health Sciences Institutional Review Board (IRB) approval (HRRC15-138), and the study was carried out in accordance with the United States Common Rule.

## Results

### Discovery single-institutional cohort

In the initial single-institutional discovery cohort from the University of New Mexico Comprehensive Cancer Center (UNMCCC), patients with intermediate-risk (Gleason Score 7; corresponding to Gleason Groups 2 and 3) organ-confined prostate cancer (n=107) met the criteria for inclusion in this study (**Table 1**). Briefly, the mean age of the cohort was 63 years (ranging from 45 to 84 years); most patients (85%) were non-Hispanic white, but Hispanic (7.5%), American Indian/Native American (2.8%), and African American (2.8%) men were also represented. All patients had final Gleason Score 7 adenocarcinoma after radical prostatectomy, with 86.9% being 3+4 = 7 (Gleason Grade Group 2) and 13.1% being 4+3 = 7 (Gleason Grade Group 3). The pathological stage distribution was as follows: 74.8% were pT2 and 25.2% were pT3a. Nineteen of the patients (17.8%) had biochemical recurrence discovered during follow-up, including one with documented local recurrence and one with documented metastases. Five patients (4.7%) were lost to follow up.

RNA extraction and qRT-PCR were successful in all microdissected tumor samples (n=107). In 24 of these cases (22.4%), we extracted RNA from benign prostatic glandular tissue away from tumor (hereafter termed 'normal prostate': qRT-PCR was successful in all cases for *PRUNE2* [n=24, 100%] and

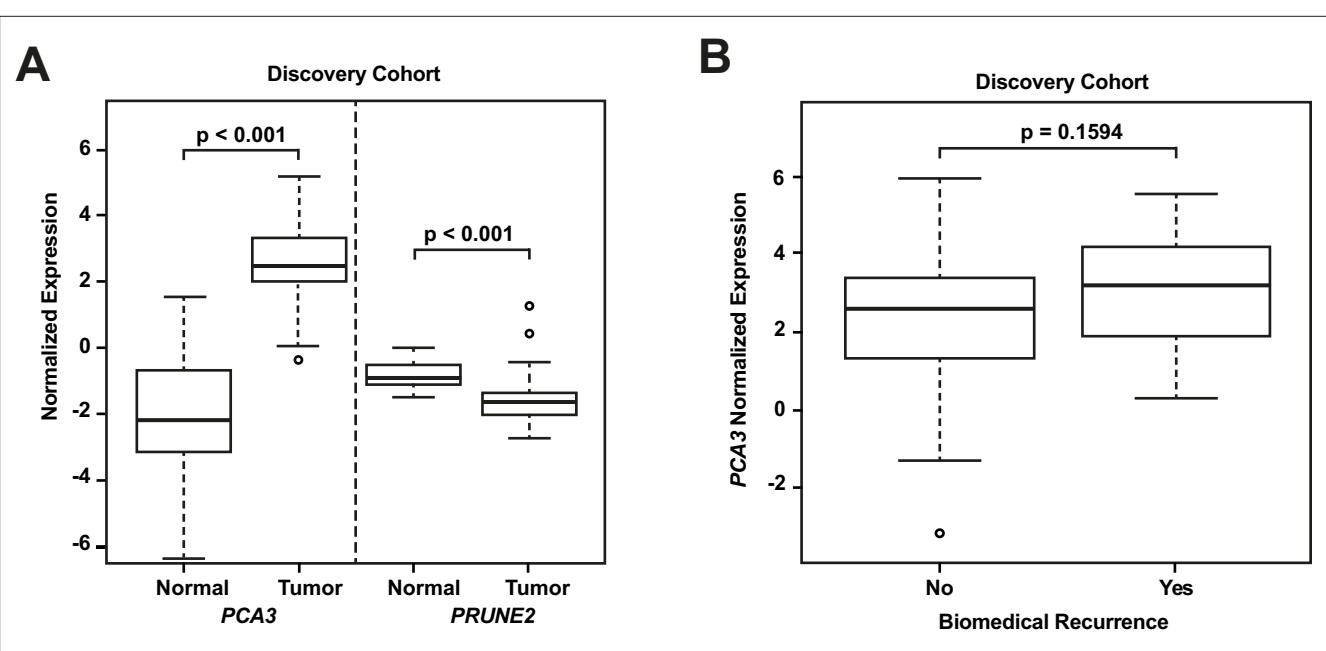

**Figure 1.** Analyses of discovery prostate cancer cohort. (**A**) *PRUNE2* and *PCA3* expression in tumor (n=107) (**Figure 1—source data 1**) and nonneoplastic (n=24 for *PRUNE2*, n=21 for *PCA3*) (**Figure 1—source data 1**) prostatic samples. Calculated values available in **Figure 1—source data 1**. (**B**) Tumor *PCA3* expression by biochemical recurrence status. *PCA3* expression in patients without versus with biochemical recurrence in the discovery cohort (n=102). No significant difference in median expression was seen in this cohort. Box plots of gene expression (normalized expression) in the discovery cohort. The horizontal line within each box represents the median value, while the box represents the interquartile range, and the whiskers extend out to 1.5 times the interquartile range. Outliers are represented by circles. p-Values are noted for the indicated comparisons.

The online version of this article includes the following source data and figure supplement(s) for figure 1:

**Source data 1.** Analyses of discovery prostate cancer cohort.

**Source data 2.** Analyses of discovery prostate cancer cohort.

**Source data 3.** Analyses of discovery prostate cancer cohort.

**Figure supplement 1.** Discovery cohort – no significant difference in tumor *PRUNE2* expression by biochemical recurrence status.

in most cases for *PCA3* [n=21, 87.5%]). Comparing *PRUNE2* and *PCA3* expression in prostatic adenocarcinoma with expression in normal prostate (all relative to endogenous housekeeping controls), we found consistent trends for both genes in multiple assays, with lower expression of *PRUNE2* in tumor as compared with normal prostate and higher expression of *PCA3* in tumor as compared with normal prostate (*Figure 1—source data 1*). These results are summarized in *Figure 1A* and as follows. Relative to controls, *PCA3* expression was significantly higher in prostatic adenocarcinoma (mean: 2.46; standard deviation: 1.28) than normal prostate (–1.99; 2.63, [p-value <0.001]). Relative to controls, *PRUNE2* expression was significantly lower in tumor (mean: –1.48; standard deviation: 0.92) than normal prostate (–0.78; 0.4, [p-value <0.001]).

We next explored the association between biochemical recurrence and tumor expression levels of *PRUNE2*, *PCA3*, and the ratio of *PRUNE2* to *PCA3* expression by using several approaches. First, we compared the gene expression values and their ratio by recurrence status. In patients who recurred compared to those who did not, we found no significant difference in mean expression values of *PRUNE2* (–1.6 to –1.58; p-value = 0.68), *PCA3* (2.98 versus 2.43; p-value = 0.16), or their ratio (–1.61 to –1.21, p-value = 0.48). The different expression levels by recurrence were not significant (*Figure 1—figure supplement 1*). Next, for *PRUNE2* expression, *PCA3* expression, and their ratio, we regrouped the cancer cases according to whether the values were greater than (deemed 'high') or less than/equal to (deemed 'low') their respective mean values. By using the Kaplan-Meier product limit methodology and the log-rank test, we found no significant associations between high or low levels and time to recurrence for *PRUNE2* expression (p-value = 0.24), *PCA3* expression (p-value = 0.22) (*Figure 2* and *Tables 2–3*), or their ratio (p-value = 0.84). As a further assessment of association between gene expression and time to biochemical recurrence, we used Cox proportional hazards modeling and found no significant associations of time to biochemical recurrence with expression of *PRUNE2* (hazard ratio [HR]: 0.97; 95% confidence interval [CI]: 0.58–1.63, [p-value = 0.91]) or *PCA3* (HR: 1.21; 95% CI: 0.91–1.6, [p-value = 0.19]), or their ratio (HR: 0.99; 95% CI: 0.92–1.1, [p-value = 0.82]). By multivariable Cox modeling, we did not find that expression of *PRUNE2*, *PCA3*, or their ratio added any additional predictive information for recurrence to that provided by clinical or pathological variables, as presented in Appendix 1 and *Appendix 1—table 1*.

## Validation/confirmation multi-institutional cohort

For the analysis of a second multi-institutional validation/confirmation and expansion prostate cancer cohort from TCGA, patient clinical data (*Table 4*) and gene expression data were available on men with prostate cancer (n=497). Gene expression data from nonneoplastic prostatic tissue (hereafter termed 'normal prostate') were also available in 52 (10.5%) of the cases. The basis for the cohort has been described previously (*Cancer Genome Atlas Research Network, 2015*). Briefly, the cohort comprised men whose ages ranged from 41 to 78 years, and who had a radical prostatectomy for primary prostate cancer. The distribution of prostate cancer grades was as follows: Gleason Score 3+3 = 6 (Grade Group 1), 9.0%; Gleason Scores 3+4 = 7 or 4+3 = 7 (Grade Groups 2 and 3), 49.7%; Gleason Score 4+4 = 8 (Grade Group 4), 12.9%; and Gleason Scores 4+5 = 9 or 5+4 = 9, 27.6%; and 5+5 = 10 (Grade Group 5), 0.8%. For pathological stage, most tumors were pT2c (33%), pT3a (32%), or pT3b (27%), with a small fraction being pT4 (2%). Data on disease recurrence were available for 429 men (86.3%), with 58 (11.7% [13.5% of those with follow-up information available]) having biochemical recurrence.

We compared gene expression levels (RSEM) in prostatic adenocarcinoma to those in normal prostate samples in the dataset from TCGA (*Figure 3A*): *PCA3* had significantly increased expression in carcinoma (median: 12.4; interquartile range [IQR]: 10.3–13.7) as compared with normal prostate (median: 6.9; IQR: 5.2–9.6, [p-value <0.001]), and *PRUNE2* showed simultaneous lower expression in carcinoma (median: 11.4; IQR: 10.7–12.0) versus normal prostate (median: 12.2; IQR: 11.8–12.6, [p-value <0.001]). As depicted in *Figure 3B*, comparing tumor gene expression across different prostate cancer pathological grades, *PCA3* expression was significantly lower in tumors with Gleason Score greater than 7 (median: 11.6; IQR: 8.4–13.4) than in tumors with Gleason Score 7 (median: 12.8; IQR: 11.3–13.8, [p-value <0.001]) or less than 7 (median: 12.5; IQR: 11.8–13.7, [p-value = 0.01]). *PRUNE2* showed a small decrease in expression in tumors with Gleason Score greater than 7 (median: 11.3; IQR: 10.4–11.9) as compared with tumors with Gleason Score 7 (median: 11.5; IQR: 10.8–12.1, [p-value = 0.014]) or less than 7 (median: 11.6; IQR: 11.0–12.1, [p-value = 0.049]). As shown in *Figure 3C*,

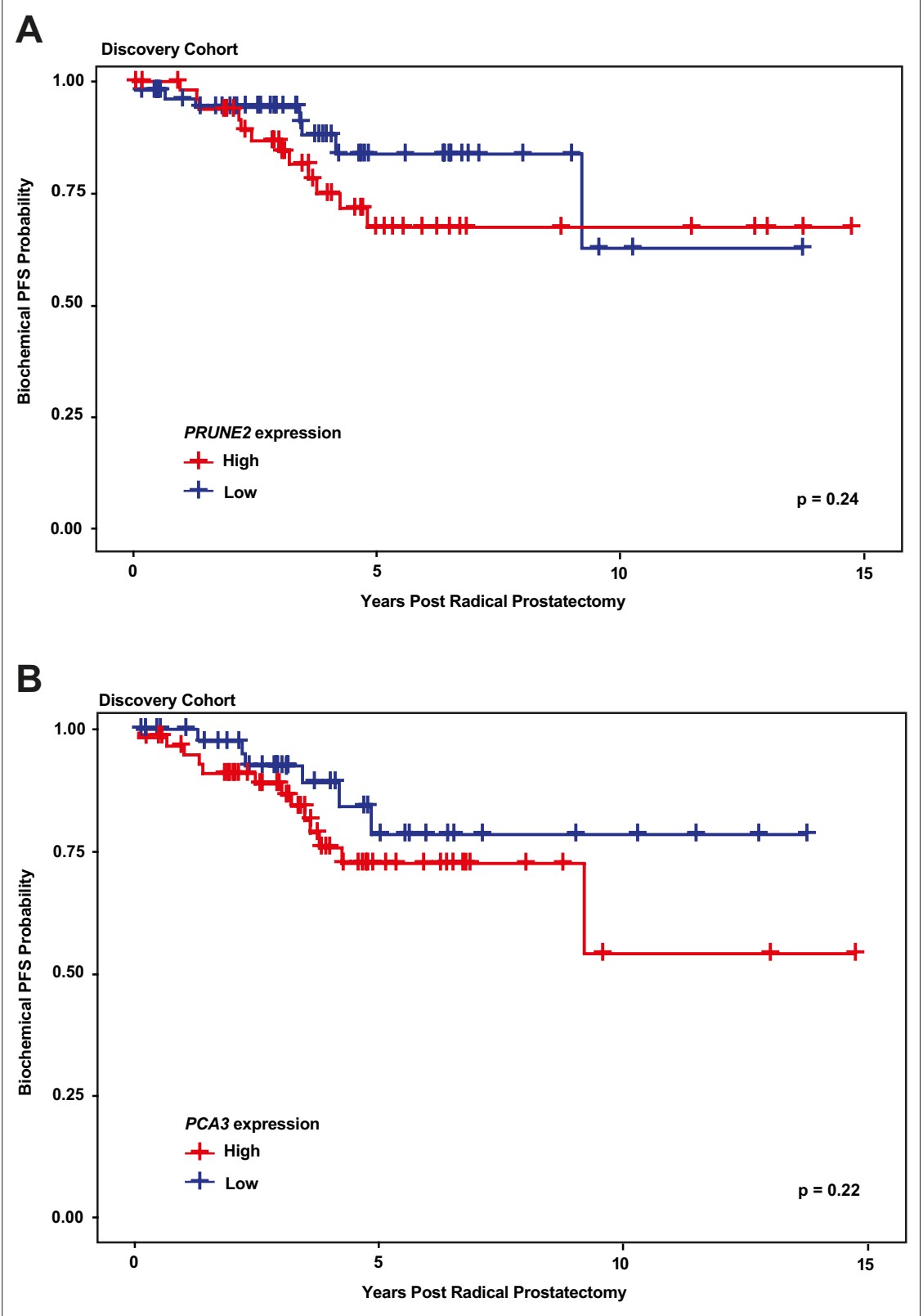

**Figure 2.** Kaplan-Meier curves illustrating time to event (biochemical recurrence) among patients post-prostatectomy from the discovery cohort, stratified by 'high' gene expression (greater than mean expression, red line) versus 'low' gene expression (less than or equal to mean expression, blue line), for (**A**) tumor *PRUNE2* expression, and (**B**) tumor *PCA3* expression. There was no significant association of high versus low expression levels and time to recurrence by log-rank testing for either *PRUNE2* or *PCA3*.

**Table 2.** Number of patients at risk over time (see *Figure 2A*).

| *PRUNE2* expression | Years post radical prostatectomy | | | |
|---|---|---|---|---|
| | 0 | 5 | 10 | 15 |
| High | 51 | 15 | 5 | 0 |
| Low | 56 | 14 | 2 | 0 |

comparing tumor gene expression across different tumor pathological stages, *PCA3* expression was higher in pT2 tumors (median: 12.6; IQR: 11.2–13.8) than in tumors that were pT3 (median: 12.2; IQR: 9.7–13.6, [p-value = 0.01]) or pT4 (median: 12.1; IQR: 9.6–12.7, [p-value = 0.61]). There was no significant difference (p-value >0.05) between *PRUNE2* expression levels between the different tumor stages: pT2 (median: 11.4; IQR: 10.7–12.1), pT3 (median: 11.3; IQR: 10.6–12.0), and pT4 (median: 11.7; IQR: 10.8–12.1). We also found that the ratio of *PCA3/PRUNE2* expression showed similar associations with Gleason Score and pathological stage as were seen with *PCA3* expression (data not shown). Overall, despite the differences in gene expression among tumor grades and stages, the median expression of *PCA3* was significantly higher in all tumor grades and stages than the expression of *PCA3* in normal prostate, and, inversely, the median expression of *PRUNE2* in all tumor grades and stages was significantly less than the expression of *PRUNE2* in normal prostate.

As shown for the discovery cohort, we also evaluated the relationship between PCA3 and PRUNE2 expression levels and recurrence status. We found that patients who had biochemical recurrence after prostatectomy had significantly lower tumor expression levels of *PCA3* (median, 11.58; IQR, 8.28–13.14) than those who did not recur (12.51; 10.64–13.71, [p-value <0.01]; *Figure 3D*). However, we did not see an association between tumor *PCA3* expression and biochemical recurrence on multivariable Cox proportional hazards modeling when adjusting for tumor grade, stage, and age at diagnosis (HR, 0.96; 95% CI, 0.87–1.04, [p-value = 0.36]), as presented in Appendix 1 and *Appendix 1—table 2*. We did not see a significant association between *PRUNE2* expression in those patients that had biochemical recurrence as compared with those patients who did not recur (*Figure 3—figure supplement 1*).

## Discussion

Here, we assessed the tumor and control adjacent normal prostatic glandular tissue expression of the lncRNA *PCA3* and the protein-coding *PRUNE2* gene in two independent retrospective cohorts of patients with primary organ-confined prostate cancer after treatment by radical prostatectomy (*Figure 4*). As compared with normal prostate, we found that prostate cancer showed consistent increased expression of *PCA3* and consistent decreased expression of *PRUNE2* in tumors across a broad range of pathological attributes (i.e., Gleason grades, scores, groups, and stages) in both patient cohorts. Although the magnitude of the change of expression between normal and tumor appears greater for *PCA3* than for *PRUNE2* in both cohorts (*Figure 1A* and *Figure 3A*), we attribute this to the reciprocal nature of the comparison, in conjunction with the very low level of normal prostatic *PCA3* expression as compared with the higher expression of *PRUNE2* in normal prostate. Overall, the findings support the mechanistic role of a tumor-specific molecular axis in which *PCA3* acts as dominant-negative oncogene and *PRUNE2* as a tumor suppressor gene in human prostate cancer and indicate that the interplay between these genes is dysregulated early in prostate cancer.

Specifically, when we compared *PCA3* expression in the validation cohort from TCGA, although average expression in all grades, stages, and groups was higher than in normal prostate, we found that among tumors there was significantly decreased *PCA3* expression in tumors with higher grades (Gleason Score >7) and in higher stages (>pT2), as compared with lower grades, stages, or groups, respectively. These paradoxical findings are consistent with several early studies (*Salagierski et al., 2010*; *Balcerczak et al., 2003*) and in particular with a recent tissue-based study of *PCA3* expression in prostate cancer (*Alshalalfa et al., 2017*). In that large cohort study, lower levels of tumor *PCA3* in both biopsy and radical prostatectomy specimens were associated with high-grade tumors, and in radical prostatectomy specimens decreased *PCA3* expression was associated with features of higher stages. Based on these results,

**Table 3.** Number of patients at risk over time (see *Figure 2B*).

| *PCA3* expression | Years post radical prostatectomy | | | |
|---|---|---|---|---|
| | 0 | 5 | 10 | 15 |
| High | 59 | 16 | 2 | 0 |
| Low | 48 | 13 | 5 | 0 |

**Table 4.** Validation cohort: clinicopathological features of the 497 patients in the prostate cancer TCGA dataset, with a total of 549 tissue samples included.

| Variable | Number | % or Mean (STD; range) |
|---|---|---|
| Age (years) | 497 | 61 (6.8; 41–78) |
| Vital status | | |
| Alive | 487 | 97.9 |
| Dead | 10 | 2.1 |
| Sample type | | |
| Primary tumor | 497 | |
| Normal (non-malignant) prostate | 52 | |
| Post-prostatectomy Gleason Grade (Grade Group) | | |
| 6 (Grade Group 1) | 45 | 9 |
| 7 (Grade Groups 2 and 3) | 247 | 49.7 |
| 8 (Grade Group 4) | 64 | 12.9 |
| 9 (Grade Group 5) | 137 | 27.6 |
| 10 (Grade Group 5) | 4 | 0.8 |
| Pathological stage | | |
| pT2a | 13 | 2.6 |
| pT2b | 10 | 2 |
| pT2c | 164 | 33 |
| pT3a | 158 | 32 |
| pT3b | 135 | 27 |
| pT4 | 10 | 2 |
| Unknown | 7 | 1.4 |
| Biochemical recurrence status | | |
| No | 371 | 74.6 |
| Yes | 58 | 11.7 |
| Unknown | 68 | 13.7 |

STD = standard deviation.

it has been proposed that *PCA3* might actually represent a differentiation marker in human prostate cancer (*Alshalalfa et al., 2017*). The finding of decreasing *PCA3* expression with increasing tumor grades and stages in both our study and others is broadly consistent with another previous study (*Reis et al., 2004*), which found that the class of antisense intronic RNAs was markedly over-represented among the top transcripts associated with tumor differentiation in human prostate cancer. The finding of an inverse association between *PCA3* expression and increasing grades and stages may also relate to links between *PCA3* expression and androgen receptor (AR) signaling and the likelihood of *PCA3* having an important role in the early steps of prostate cancer carcinogenesis, with a reduced role when the disease is more advanced. Indeed, previous work by our own group and by others indicates that *PCA3* is upregulated by AR signaling (*Teixeira et al., 2017*; *Ferreira et al., 2012*; *Salameh et al., 2015*), and that *PCA3* is also involved in modulating AR signaling (*Ferreira et al., 2012*; *Lemos et al., 2016*). Interestingly, it has also been shown in vitro that *PCA3* silencing sensitizes prostate cancer cells to enzalutamide-induced decreased cell growth (*Özgür et al., 2017*). *Alshalalfa et al., 2017*, suggest that because low pretreatment serum testosterone levels are associated with diseases with higher grades and stages, and because of the relationship between AR signaling and *PCA3* expression,

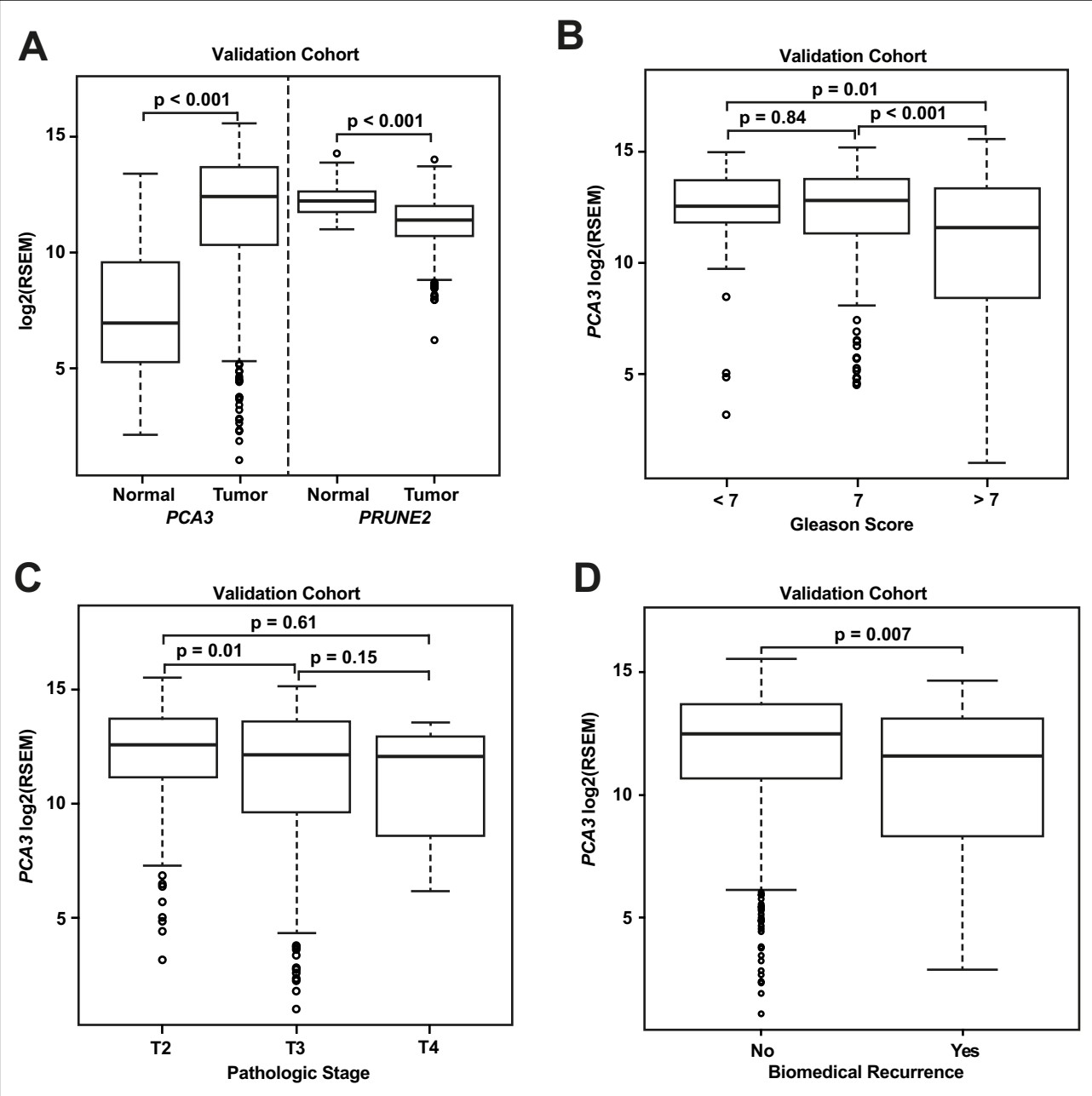

**Figure 3.** Analyses of TCGA prostate cancer validation cohort. (**A**) *PCA3* and *PRUNE2* expression in nonneoplastic prostatic glandular tissue and in prostatic adenocarcinoma in the TCGA cohort illustrating consistent gene expression differences between tumor (n=497) and nonneoplastic (n=52) prostate in both cohorts. (**B, C**) *PCA3* expression in the cohort from TCGA (n=497) across Gleason grades (**B**) showing lower expression in higher grade (>7) tumors and across tumor stages (**C**) showing lower expression in higher stage tumors. All illustrated tumor grades and stages show higher expression than nonneoplastic prostatic epithelium. (**D**) *PCA3* expression in patients without versus with biochemical recurrence in the TCGA cohort (n=429). In the TCGA cohort, lower *PCA3* median expression was associated with biochemical recurrence. Box plots of gene expression in the TCGA cohort is reported as log₂RSEM data. The horizontal line within each box represents the median value, while the box represents the interquartile range, and the whiskers extend out to 1.5 times the interquartile range. Outliers are represented by circles. p-Values are noted for the indicated comparisons. RSEM: RNA-Seq by Expectation-Maximization.

The online version of this article includes the following figure supplement(s) for figure 3:

**Figure supplement 1.** Validation cohort – no significant difference in tumor *PRUNE2* expression by biochemical recurrence status.

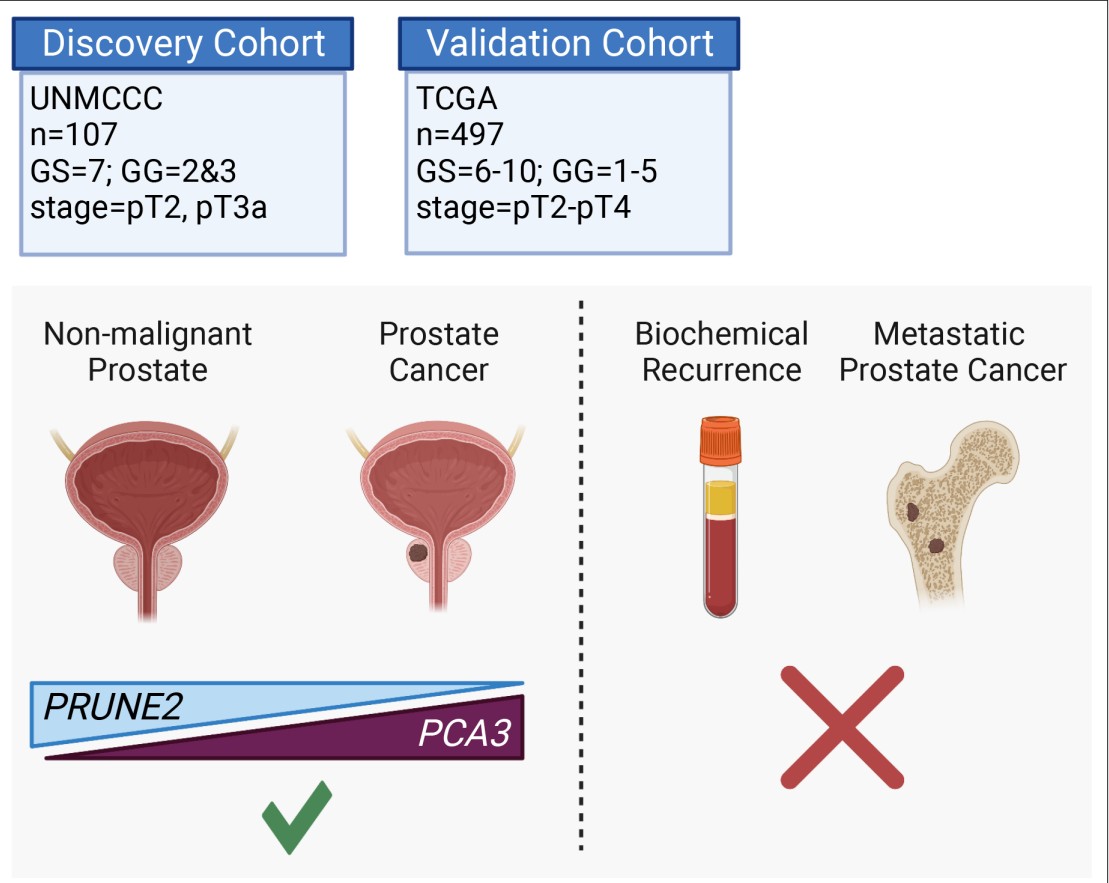

**Figure 4.** Graphical summary of the analyses. *PCA3* and *PRUNE2* gene expression relationship in paired prostate cancer and adjacent normal prostate was analyzed in two independent retrospective cohorts of clinically annotated cases post-radical prostatectomy: a single-institutional discovery cohort (n=107; *Figure 4—figure supplement 1*) and a multi-institutional validation cohort (n=497). We also serially examined clinical/pathological variables including time to disease recurrence. Created with BioRender.com.

The online version of this article includes the following figure supplement(s) for figure 4:

**Figure supplement 1.** Flow diagram of discovery cohort selection criteria and clinicopathological characteristics available.

therefore lower *PCA3* expression may reflect the lower serum testosterone in these patients. However, we do not have any data on the pretreatment serum concentration of testosterone and other androgens, and we are not able to test that hypothesis in this study.

Because prostate cancers, especially Gleason Score 7 (Grade Groups 2 and 3) tumors, are quite frequent (about half of the total cases) and show divergent clinical behavior, there is great interest in developing prognostic biomarkers for risk stratification. Studies on the association of *PCA3* expression levels with outcome and prognosis show conflicting results (*Loeb and Partin, 2011*), and unlike this present study, most prior reports are based on urinary *PCA3* expression (*Loeb et al., 2015*; *Lemos et al., 2019*; *Fenstermaker et al., 2017*). Our exploration of the validation cohort from TCGA, which comprised a wide spectrum of tumor grades and stages, revealed an association between lower levels of tumor *PCA3* expression and biochemical recurrence; however, this association was not found after taking grade and stage into account. This finding makes sense, as increasing grade and stage are both variables that are associated with lower *PCA3* expression. In their tissue-based cohort, *Alshalalfa et al., 2017*, also found an association between low *PCA3* levels and adverse outcomes, including biochemical recurrence, metastasis, and prostate cancer-specific mortality; however, it is not clear whether such findings are independent of clinical and pathological variables (such as Gleason grade, stage, and group), as a multivariable analysis was not reported. Nevertheless, the demonstration of an (unadjusted) association between *PCA3* levels and outcome may have potential relevance in the liquid biopsy setting. For the discovery cohort of patients, we selected organ-confined, intermediate-risk

tumors (Gleason Grade Groups 2 and 3, with tumor stages pT2 and pT3) where prognostic information might be expected to be most helpful clinically, to test for an association with outcome. We did not see any association between tumor *PCA3* expression and biochemical recurrence in this particular grade and stage setting.

*PRUNE2*, a human homolog of the *Drosophila prune* gene, encodes for a protein with BCH, DHHA2, and PPX1 functional domains (*Ferreira et al., 2012*). The BCH domain can inhibit the Rho family of proteins, small GTPases with roles in cell transformation, migration and metastasis, and cell cycle progression (*Clarke et al., 2009*; *Iwama et al., 2011*). Evidence is accumulating that *PRUNE2* might act as a tumor suppressor gene. Loss-of-function mutations have been described in several tumor types, including germline and somatic mutations in parathyroid cancer (*Yu et al., 2015*) and somatic mutations in solid papillary carcinoma (*Alsadoun et al., 2018*), while high expression of PRUNE2 protein correlates with favorable prognosis in neuroblastoma (*Machida et al., 2006*). Others have shown evidence of inactivating *PRUNE2* mutations in Merkel cell carcinoma (*Harms et al., 2015*) and that the restoration of downregulated PRUNE2 in oral cancer suppresses tumor cell migration (*Su et al., 2021*), further supporting the role of *PRUNE2* as a tumor suppressor. In prostate cancer, the evidence is limited and controversial: an early report found that *PRUNE2* expression was upregulated in prostate cancer and metastases in a small number of samples, and was androgen-inducible in prostate cancer cells (*Clarke et al., 2009*). However, a subsequent study on a larger number of samples found that *PRUNE2* expression either decreased or did not increase in aggressive prostate cancer, and that *PRUNE2* expression was not androgen-inducible (*Salagierski et al., 2010*). While this work was under external peer-review, Cardoso et al. have shown that *PRUNE2* is a prostate cancer predisposition gene, which is consistent with our results and interpretations (*Cardoso et al., 2022*).

Altogether, the findings in the current study provide additional support for our previous findings (*Salameh et al., 2015*) that *PRUNE2* acts as a functional tumor suppressor gene in human prostate cancer. Here, we described consistently lower expression of *PRUNE2* in prostate cancers of all grades and stages as compared to normal prostate. The findings in our present study are also consistent with the negative regulation of *PRUNE2* by *PCA3* in prostate cancer. We found no significant differences in *PRUNE2* expression across tumor stage, and only a small decrease in expression with increasing tumor grade, suggesting that loss of *PRUNE2* tumor suppressor activity is an early molecular event in prostate cancer. We are not aware of any prior reports of the prognostic significance of tumor *PRUNE2* expression in prostate cancer but, at least in this retrospective study of two independent prostate cancer patient cohorts, we did not find any association between *PRUNE2* expression and biochemical outcomes.

Strengths of this study include that broadly consistent findings were described in the two independent well-characterized clinically annotated primary prostate cancer cohorts used for analysis, and that the findings were robust across multiple assays in the discovery patient cohort and between the different methods of measurement of gene expression used in the two cohorts. The assessment of *PCA3* expression directly and specifically in tissue (as opposed to urine) is a novelty and a strength as our primary goal was the study of the *PRUNE2/PCA3* regulatory axis in human prostate cancer. We reasoned that the study of tissue expression is likely more informative of tumor biology than traditional urinalysis, not least of all because urinary expression, though very well characterized, could by subject to potential confounding issues such as RNA stability in urine or the contribution of differential urinary shedding. However, from the standpoint of assessment of prognostic information, a drawback of analyzing tissue *PCA3* expression is that the results are not directly comparable to the multiple previous studies that measured urinary *PCA3* scores and ultimately led to FDA and EMA approval for clinical applications in the US and EU. Moreover, while we did find consistent findings with a large tissue cohort study relating *PCA3* expression and biochemical recurrence (*Alshalalfa et al., 2017*), the analysis presented here was limited in its ability to unequivocally determine the prognostic value of *PCA3* and *PRUNE2* expression as the overall proportion of patients with biochemical recurrences was relatively low. Finally, we were not able to fully address the relationship of reciprocal gene expression of *PCA3* and *PRUNE2* to the outcomes of metastases and prostate cancer-specific deaths, again due to the relative paucity of these events.

In conclusion, we found consistent upregulation of *PCA3* and downregulation of *PRUNE2* in prostate cancer as compared with normal prostate in two retrospective and independent patient cohorts (summarized in *Figure 4*, *Figure 4—figure supplement 1*), supporting that *PCA3* and *PRUNE2*

function as an oncogene and a tumor suppressor gene, respectively, in human prostate cancer. The inverse correlation of *PCA3* and *PRUNE2* expression is consistent with our prior findings of a functional interplay between the two genes as part of a unique regulatory unit functioning at a single genetic locus in prostate cancer cells with *PCA3* negatively downregulating *PRUNE2* expression (*Salameh et al., 2015*). The mechanistic dysregulation of *PCA3* and *PRUNE2* is observed across the spectrum of tumor grades and stages, suggesting that this is an early and stable molecular event in prostate cancer. On the other hand, we have not detected any regulatory effects of *PRUNE2/PCA3* in late genetic events such as prostate cancer progressing to biochemical recurrence, which includes the development of local tumor recurrence and/or the development of metastatic disease. The findings presented here represent additional evidence for the functional reciprocal co-regulation of *PCA3* and *PRUNE2* in the setting of early tumorigenesis but not in late events in human prostate cancer. Taken together along with the well-documented specificity of *PCA3* overexpression, our findings establish the *PCA3/PRUNE2* regulatory axis as an attractive early molecular target candidate for intervention in the therapy of human prostate cancer.

## Acknowledgements

We received support from the Human Tissue Repository and Tissue Analysis Shared Resource from the Department of Pathology of the University of New Mexico School of Medicine and a pilot award from the UNMCCC. RP and WA were supported by awards from the Levy-Longenbaugh Donor-Advised Fund and the Prostate Cancer Foundation. EDN reports research fellowship support from the Brazilian National Council for Scientific and Technological Development (CNPq), Brazil, and the Associação Beneficente Alzira Denise Hertzog Silva (ABADHS), Brazil. This work has been funded in part by the NCI Cancer Center Support Grants (CCSG; P30) to the UNMCCC (CA118100) and the Rutgers Cancer Institute of New Jersey (CA072720). The results shown here are in part based upon data generated by the TCGA Research Network: https://www.cancer.gov/tcga. These data have been presented in part at the 2017 American Society of Clinical Oncology Annual Meeting.

## Additional information

### Competing interests

Diana N Nunes: The University of New Mexico filed patent applications on PRUNE2- related technology, for which Diana Nunes was an inventor (inventors: DNN, EDN, RP, and WA). Those applications were briefly optioned by MBrace Therapeutics, but the applications have since been abandoned and the agreements terminated. No payments were made to Diana Nunes, and the author has no other competing interests to declare. Emmanuel Dias-Neto: The University of New Mexico filed patent applications on PRUNE2- related technology, for which Emmanuel Dias-Neto was an inventor (inventors: DNN, EDN, RP, and WA). Those applications were briefly optioned by MBrace Therapeutics, but the applications have since been abandoned and the agreement terminated. No payments were made to Emmanuel Dias-Neto, and the author has no other competing interests to declare. Isan Chen: serves as the Chief Executive Officer of MBrace Therapeutics. Mbrace did not provide financial support for the present work. Webster K Cavenee: is a founder and shareholder of Interleukin Combinatorial Therapies, Inc, InVaMet, Inc, and io9, LLC; none of these companies provided funds or participated in the present work. These arrangements are managed in accordance with the established institutional conflict of interest policies for the respective institution. The author received support for attending the Aspen Cancer Conference, and participated in a Leadership or fiduciary role. The author holds a Leadership or fiduciary role at Genetron Health for which they receive board fees, and are on the Board of Directors for the GBM AGILE Clinical Trial. The author has no other competing interests to declare. Renata Pasqualini, Wadih Arap: Reviewing editor, eLife. The other authors declare that no competing interests exist.

## Funding

| Funder | Grant reference number | Author |
| --- | --- | --- |
| National Cancer Institute | P30CA118100 | Richard C Lauer |
| National Cancer Institute | P30CA072720 | Renata Pasqualini Wadih Arap |
| Levy-Longenbaugh Donor-Advised Fund | | Renata Pasqualini Wadih Arap |
| Prostate Cancer Foundation | | Renata Pasqualini Wadih Arap |
| Foundation for Scientific and Technological Development in Health | | Emmanuel Dias-Neto |
| Associação Beneficente Alzira Denise Hertzog Silva | | Emmanuel Dias-Neto |
| University of New Mexico | Comprehensive Cancer Center Pilot Award | Richard C Lauer |

The funders had no role in study design, data collection and interpretation, or the decision to submit the work for publication.

## Author contributions

Richard C Lauer, Conceptualization, Data curation, Formal analysis, Supervision, Funding acquisition, Investigation, Writing – original draft; Marc Barry, Conceptualization, Data curation, Formal analysis, Supervision, Investigation, Writing – original draft; Tracey L Smith, Andrew Maltez Thomas, Jin Wu, Ruofei Du, Ji-Hyun Lee, Arpit Rao, Andrey S Dobroff, Marco A Arap, Diana N Nunes, Israel T Silva, Data curation, Formal analysis, Investigation, Writing – review and editing; Emmanuel Dias-Neto, Dennis J McCance, Conceptualization, Data curation, Formal analysis, Funding acquisition, Investigation, Writing – original draft; Isan Chen, Webster K Cavenee, Formal analysis, Writing – review and editing; Renata Pasqualini, Wadih Arap, Conceptualization, Formal analysis, Supervision, Funding acquisition, Writing – original draft

## Author ORCIDs

Andrew Maltez Thomas http://orcid.org/0000-0001-5789-3354
Ji-Hyun Lee http://orcid.org/0000-0001-6420-5150
Wadih Arap http://orcid.org/0000-0002-8686-4584

## Ethics

Human subjects: For the discovery cohort, there was University of New Mexico Health Sciences Institutional Review Board (IRB) approval (HRRC15-138), and the study was carried out in accordance with the United States Common Rule. As the discovery cohort involved secondary use of archival biospecimens, the IRB waived the requirement for informed consent .

## Decision letter and Author response

Decision letter https://doi.org/10.7554/eLife.81929.sa1
Author response https://doi.org/10.7554/eLife.81929.sa2

# Additional files

## Supplementary files
- MDAR checklist
- Reporting standard 1. STROBE Checklist.
- Source code 1. R code and SAS code of descriptive statistics.

## Data availability

For the discovery cohort, all data generated or analyzed are included in the manuscript and source data files, except for patient-level ethnicity data. Patient-level ethnicity data is not included due to the potential for identifiability. However detailed summary ethnicity data is presented in the manuscript

and in Table 1. Requests to access the patient level ethnicity data should be directed to the corresponding author with a project proposal. Source codes are also available in the supplemental source code file. For the Validation Cohort, clinicopathological patient characteristics and gene level transcription data from The Cancer Genome Atlas (TCGA) were accessed from the UCSC Xena Resource.

The following previously published dataset was used:

| Author(s) | Year | Dataset title | Dataset URL | Database and Identifier |
|---|---|---|---|---|
| University of North Carolina TCGA genome characterization center | 2017 | TCGA prostate adenocarcinoma (PRAD) gene expression by RNAseq (polyA+ IlluminaHiSeq) | https://xenabrowser.net/datapages/?dataset=TCGA.PRAD.sampleMap%2FHiSeqV2&host=https%3A%2F%2Ftcga.xenahubs.net&removeHub=https%3A%2F%2Fxena.treehouse.gi.ucsc.edu%3A443 | TCGA Prostate Cancer (PRAD) TCGA.PRAD.sampleMap/HiSeqV2, 20,531 |

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

# Appendix 1

## Results

In the discovery cohort, to further assess the association between time to biochemical recurrence and the gene expression of *PRUNE2* and *PCA3* and the ratio of their expressions, and to take into account possible confounding by clinicopathological variables, we performed multivariate Cox proportional hazards regression modeling (raw data available in Appendix tables and code). As there may be possible correlations between tumor Gleason Grade (GG) and tumor pathological stage (PS), we created a composite categorical variable ('GG-PS') representing the four possibilities in the discovery cohort: 3+4/pT2, 4+3/pT2, 3+4/pT3a, 4+3/pT3a. We then used two approaches to model outcome.

In the first approach, we fit four different Cox models. The explanatory variables in these models were as follows: model 1 ('GG_PS model') – GG_PS only; model 2 ('*PRUNE2* model') – *PRUNE2* expression, age, race, GG_PS, and interaction between GG_PS and *PRUNE2* expression; model 3 ('*PCA3* model') – *PCA3* expression, age, race, GG_PS, and interaction between GG_PS and *PCA3* expression; and model 4 ('Ratio model') – *PRUNE2/PCA3* ratio, age, race, GG_PS, and interaction between GG_PS and the ratio. The goodness-of-fit of the models were compared using the Akaike information criterion (AIC), and the results are summarized in *Appendix 1—table 1*. As a lower value of AIC indicates a better association fit, the model comparison indicates that the model with GG_PS only (model 1) represents the best fit for the data, and does not suggest that the expression of *PRUNE2*, *PCA3*, or their ratio adds to the ability of pathology grade and stage to predict biochemical recurrence.

**Appendix 1—table 1.** Discovery cohort multivariable model.

| Model | Akaike information criterion score |
| --- | --- |
| PS_GG model | 157.2 |
| *PRUNE2* model | 162.9 |
| *PCA3* model | 166.2 |
| Ratio model | 162.2 |

As a second multivariable approach to an assessment of the association of time to biochemical recurrence and gene expression, we created a multivariable model, including the following variables: expression of *PRUNE2*, expression of *PCA3*, age, race, the composite variable GG_PS, interaction between *PRUNE2* expression and GG_PS, and interaction between *PCA3* expression and GG_PS. In this case, a stepwise selection algorithm was used for model selection, and only the model with Gleason score and pathological stage (GG_PS) was selected, as none of the other variables had a p-value less than the specified significance level of 0.25 (data not shown).

In the TCGA cohort, we used Cox proportional hazards modeling to assess the association of tumor PCA3 expression, adjusting for tumor grade, tumor stage, and age at diagnosis. The results are summarized in *Appendix 1—table 2*. Briefly, on multivariable modeling, there was an association between tumor grade and stage with recurrence, we did not find that tumor *PCA3* expression was associated with biochemical recurrence (HR, 0.96; 95% CI 0.87–1.04, [p-value = 0.36]).

**Appendix 1—table 2.** Validation cohort multivariable Cox model.

| | Hazard ratio (HR) | 95% CI | p-Value |
| --- | --- | --- | --- |
| *PCA3* | 0.963 | 0.888–1.044 | 0.36 |
| Gleason Grade | 1.558 | 1.147–2.117 | <0.01 |
| Pathological stage T3 | 3.596 | 1.360–9.512 | <0.01 |
| Pathological stage T4 | 1.860 | 0.206–16.82 | 0.58 |
| Age at diagnosis | 1.000 | 0.960–1.043 | 0.99 |

## Methods

### Statistical analysis for quantifying the expression of PCA3 and PRUNE2

There were combinations of assays and control genes used for quantifying the expression of *PCA3* and *PRUNE2* in this study. Explicitly, there were nine duplex mixes for *PRUNE2*: PR1C1, PR1C2, PR1C3, PR2C1, PR2C2, PR2C3, PR3C1, PR3C2, PR3C3; and six duplex mixes for *PCA3*: PC1C1, PC1C2, PC1C3, PC2C1, PC2C2, PC3C3, where the first three letters denote an assay and last two letters denote a control gene being used in a particular run. For example, PC2C2 denotes the second assay for *PCA3* (Hs03462121_m1, detailed in Methods) and the second endogenous control gene (Hs02800695_m1, detailed in Methods) were used for that specific experiment.

$C_T$ is to denote the logarithmic number of PCR cycle when the fluorescent signal passes a threshold value. Let $\Delta C_T = C_{T \text{ study gene}} - C_{T \text{ control gene}}$ and we had $-\Delta C_T$ to quantify the gene expression (relative to a control gene), resulting in a positive value meaning an upregulated gene's expression.

The experiment was completed three times for each gene duplex mix, for example, we have three data points of PC2C2 measure for a tumor sample. The median of the three $-\Delta C_T$ values is summarized to estimate the gene expression of a particular gene duplex mix. We then looked at both mean and median of nine estimates for *PRUNE2* and six estimates for *PCA3*, separately (data not shown). We did not see any significant difference utilizing mean or median in this or subsequent analyses.

