## [Editor Report]

In this manuscript, the upregulation of PCA3 and downregulation of PRUNE2 in prostate cancer as compared with normal prostate were first discovered. The conclusions are reached through two large-scale clinical-derived cohorts, supporting that PCA3 and PRUNE2 function as an oncogene and a tumor suppressor gene, respectively. The major findings opened up a novel avenue for prostate cancer treatment.

---

## [Decision Letter]

**Decision letter after peer review:**

Thank you for submitting your article "Dysregulation of the PRUNE2/PCA3 genetic axis in human prostate cancer: From experimental discovery to validation in two independent patient cohorts" for consideration by *eLife*. Your article has been reviewed by 3 peer reviewers, and the evaluation has been overseen by a Reviewing Editor and Caigang Liu as the Senior Editor. The reviewers have opted to remain anonymous.

Essential revisions:

1) Knockdown and/or overexpression would be preferred to evaluate the role of PCA3 or PRUNE2 in prostate cancer cells.

2) Some issues are expected to be mentioned in the Discussion, including the reason that increase in PCA3 decrease in Figure 1A appears to be quite larger than the decrease in PRUNE2, and the possible utility of liquid biopsy for PCA3 beyond tissue analysis.

3) The finding that PCA3 may be related to low serum testosterone in these patients needs further data to support it.

*Reviewer #1 (Recommendations for the authors):*

The conclusions of this paper are mostly well supported by data, but some analyses need to be clarified and extended.

Suggestion:

Knockdown and/or overexpression should be applied to evaluate the role of PCA3 or PRUNE2 in prostate cancer cells.

*Reviewer #3 (Recommendations for the authors):*

In the present manuscript, Lauer and colleagues analyzed the tumor and adjacent normal prostatic glandular tissue expression of the LncRNA PCA3 and PRUNE2 in two independent retrospective cohorts of patients. The authors observed PCA3 up-regulation and PRUNE2 down-regulation in prostate cancer compared with the adjacent normal prostate across all tumor grades and stages. The authors also found there was no association between tumor PCA3 or PRUNE2 expression and biochemical recurrence. The findings provided evidence for the functional reciprocal co-regulation of PCA3 and PRUNE2 in the setting of early tumorigenesis but not in late events in human prostate cancer. Overall, this study suggests the PCA3/PRUNE2 regulatory axis as an attractive early molecular target candidate to guide the choice of appropriate treatment for human prostate cancer. The strengths of this study are that the authors found consistent findings by analyzing two independent well-characterized clinically annotated primary prostate cancer cohorts. PCA3 expression assessed by using tissue, not urine is also novel. However, the authors propose that the low expression of PCA3 may be related to low serum testosterone in these patients. It would be better if the authors have further data to support it.

---

## [Author Response]

Essential revisions:1) Knockdown and/or overexpression would be preferred to evaluate the role of PCA3 or PRUNE2 in prostate cancer cells.

Thank you for this insightful suggestion. Indeed, we have previously performed and reported these experiments (Salameh et al. *Proc. Natl. Acad. Sci. USA* 112(27)8403-8408, 2015): We assessed the functional role of the *PCA3/PRUNE2* regulatory axis by using lentiviral constructs in prostate epithelial cells in vitro. We found that *PRUNE2* ectopic expression or *PCA3* silencing decreased proliferation and transformation, while ectopic *PCA3* expression or *PRUNE2* silencing led to an increase in cell proliferation and transformation. Together with the findings in the current work, those in vitro findings support the working hypotheses that *PCA3* acts as an oncogene, and *PRUNE2* as a tumor suppressor in prostate cancer. We have now reinforced the data mentioned above in the revised manuscript (pp. 6-7, 18-19).

2) Some issues are expected to be mentioned in the Discussion, including the reason that increase in PCA3 decrease in Figure 1A appears to be quite larger than the decrease in PRUNE2, and the possible utility of liquid biopsy for PCA3 beyond tissue analysis.

We appreciate the useful suggestion towards the development liquid biopsy applications. We have now addressed this future tantalizing possibility in the Discussion of the revised manuscript (p. 20). We have also addressed the intriguing seemingly variable magnitudes of *PCA3* and *PRUNE2* in the Discussion of the revised manuscript (p. 18).

3) The finding that PCA3 may be related to low serum testosterone in these patients needs further data to support it.

We fully agree that further data will shed light on this question. As such, we have now clarified in the Discussion of the revised manuscript that the hypothesis originally put forward by Alshalalfa et al. (reference 19 of the revised manuscript) to explain the inverse association between *PCA3* levels and increasing grades and stages of prostatic cancer is an interesting open question to be addressed in further prospective clinical/epidemiological studies. In these retrospective cohorts, we do not have the data to directly assess that intriguing clinical hypothesis. However, one should also note that we have previously performed mechanistic experiments with testosterone analogues that yielded internally consistent results with our original hypothesis (Salameh et al. *Proc. Natl. Acad. Sci. USA* 112(27)8403-8408, 2015). Those aspects are now mentioned in the revised manuscript (p. 19 and reference 19).

Reviewer #1 (Recommendations for the authors):The conclusions of this paper are mostly well supported by data, but some analyses need to be clarified and extended.Suggestion:Knockdown and/or overexpression should be applied to evaluate the role of PCA3 or PRUNE2 in prostate cancer cells.

We appreciate the careful review of the manuscript by Referee #1. As indicated above, we have previously performed these experiments (Salameh et al. *Proc Natl Acad Sci USA* 112(27)8403-8408, 2015); specifically, we experimentally assessed the functional role of the *PCA3/PRUNE2* regulatory axis by using lentiviral constructs in prostate epithelial cells in vitro to show that *PRUNE2* ectopic expression or *PCA3* silencing decreased proliferation and transformation, while ectopic *PCA3* expression or *PRUNE2* silencing led to an increase in cell proliferation and transformation. Those aspects are now discussed (pp. 6-7, 18-19 and reference 8) of the revised manuscript.

Reviewer #3 (Recommendations for the authors):In the present manuscript, Lauer and colleagues analyzed the tumor and adjacent normal prostatic glandular tissue expression of the LncRNA PCA3 and PRUNE2 in two independent retrospective cohorts of patients. The authors observed PCA3 up-regulation and PRUNE2 down-regulation in prostate cancer compared with the adjacent normal prostate across all tumor grades and stages. The authors also found there was no association between tumor PCA3 or PRUNE2 expression and biochemical recurrence. The findings provided evidence for the functional reciprocal co-regulation of PCA3 and PRUNE2 in the setting of early tumorigenesis but not in late events in human prostate cancer. Overall, this study suggests the PCA3/PRUNE2 regulatory axis as an attractive early molecular target candidate to guide the choice of appropriate treatment for human prostate cancer. The strengths of this study are that the authors found consistent findings by analyzing two independent well-characterized clinically annotated primary prostate cancer cohorts. PCA3 expression assessed by using tissue, not urine is also novel. However, the authors propose that the low expression of PCA3 may be related to low serum testosterone in these patients. It would be better if the authors have further data to support it.

We must agree with all these fine points. First, PCA3-based quantitative PCR is currently approved by the FDA and the EMA for clinical use, but only in urine assays. Thus, potential applications in patient-derived serum or tissue represent yet another novel aspect of the work reported here. Moreover, we have now expanded on the interesting hypothesis by Alshalalfa et al. (reference 19 of the revised manuscript) to explain the inverse association between PCA3 levels and increasing grades and stages of prostatic adenocarcinoma. Finally, while we do not have data to directly assess it in the cohorts in the retrospective cohorts presented here, one should note that we have previously performed mechanistic experiments with testosterone analogues that yielded internally consistent results with the original hypothesis (Salameh et al. *Proc. Natl. Acad. Sci. USA* 112(27)8403-8408, 2015). Those aspects are now mentioned in the revised manuscript (p. 19 and references 8 and 19).